 

# A common directional tuning mechanism of *Drosophila* motion-sensing neurons in the ON and in the OFF pathway

**Juergen Haag\*, Abhishek Mishra, Alexander Borst**

Max-Planck-Institute of Neurobiology, Martinsried, Germany

**Abstract** In the fruit fly optic lobe, T4 and T5 cells represent the first direction-selective neurons, with T4 cells responding selectively to moving brightness increments (ON) and T5 cells to brightness decrements (OFF). Both T4 and T5 cells comprise four subtypes with directional tuning to one of the four cardinal directions. We had previously found that upward-sensitive T4 cells implement both preferred direction enhancement and null direction suppression (Haag et al., 2016). Here, we asked whether this mechanism generalizes to OFF-selective T5 cells and to all four subtypes of both cell classes. We found that all four subtypes of both T4 and T5 cells implement both mechanisms, that is preferred direction enhancement and null direction inhibition, on opposing sides of their receptive fields. This gives rise to the high degree of direction selectivity observed in both T4 and T5 cells within each subpopulation.
DOI: https://doi.org/10.7554/eLife.29044.001

## Introduction

The direction of visual motion is crucial for fundamental behaviors such as mate detection, prey capture, predator avoidance and visual navigation. This important visual cue, however, is not explicitly encoded at the output of a single photoreceptor but rather has to be computed by subsequent neural circuits. In order to extract local, directional information from moving images, mainly two competing algorithmic models of motion detectors have been proposed (*Figure 1a,b*). Both models implement a delay-and-compare mechanism where two input signals from neighboring image pixels interact in a nonlinear way after one of them has been delayed with respect to the other. This leads to an output that is larger for motion along one, the so-called 'preferred' direction than for the opposite, the so-called 'null' direction. Both models differ, however, by the type of non-linearity employed and the location of the delay. In the Hassenstein-Reichardt detector (*Figure 1a*), the delay is on the preferred side, that is where a preferred direction stimulus is entering the receptive field of the detector, and the non-linearity is excitatory. This leads to an enhancement of signals moving in the preferred direction (*Hassenstein and Reichardt, 1956*). In the Barlow-Levick detector (*Figure 1b*), the delay is on the null side, that is where a null direction stimulus is entering the receptive field of the detector, and the nonlinearity is inhibitory. This leads to a suppression of signals moving in the null direction (*Barlow and Levick, 1965*). While the predictions of both models concerning the responses to smooth grating motion are identical, apparent motion stimuli lend themselves well to discriminate between them (*Egelhaaf and Borst, 1992*; *Eichner et al., 2011*). Instead of moving an object smoothly across the image plane, an apparent motion stimulus consists of a bright or dark bar or spot that is abruptly jumped from one location to an adjacent one. Comparing the responses of directional neurons to the sequence with the sum of the responses to each individual stimulus presentation ('linear expectation') allows one to calculate the nonlinear response component as the difference between the sequence response and the linear expectation. If this nonlinear response component is positive for sequences along the preferred direction, and zero for

\*For correspondence:
haag@neuro.mpg.de

sequences along the null direction, a preferred direction enhancement is at work, supporting the Hassenstein-Reichardt model (*Figure 1a*). If the nonlinear response component is zero for sequences along the preferred direction and negative for sequences along the null direction, a null direction suppression is at work, supporting the Barlow-Levick model (*Figure 1b*). In the following, we will apply this approach in order to investigate which of the two mechanisms is at work in primary motion-sensitive neurons of the fruit fly *Drosophila*.

In *Drosophila*, visual signals are processed in the optic lobe, a brain area comprised of the lamina, medulla, lobula, and lobula plate, each arranged in a columnar, retinotopic fashion (for review, see: *Borst, 2014*; *Behnia and Desplan, 2015*). In striking parallel to the vertebrate retina (*Borst and Helmstaedter, 2015*), the direction of visual motion is computed within the optic lobe separately in parallel ON and OFF motion pathways (*Joesch et al., 2010*; *Reiff et al., 2010*; *Eichner et al., 2011*; *Joesch et al., 2013*). Anatomically, these two pathways split at the level of the lamina (*Bausenwein et al., 1992*; *Rister et al., 2007*) and lead, via a set of various intrinsic medulla and transmedulla interneurons, onto the dendrites of T4 and T5 cells, respectively. First described by Golgi staining (*Cajal and Sanchez, 1915*; *Strausfeld, 1976*; *Strausfeld and Lee, 1991*; *Fischbach and Dittrich, 1989*), T4 cells extend their dendrites in the most proximal layer of the medulla, while the dendrites of T5 cells are located in the inner-most layer of the lobula. There exist generally four T4 and 4 T5 cells per column (*Mauss et al., 2014*). The four subtypes of T4 cells respond selectively to brightness increments moving along one of the four cardinal directions, the four subtypes of T5 cells selectively to brightness decrements moving along the same four cardinal directions as T4 cells (*Maisak et al., 2013*). According to their preferred direction, T4 and T5 cells project into one of the four lobula plate layers (layer 1, most frontal: front-to-back; layer 2: back-to-front; layer 3: upward; layer 4, most posterior: downward; *Maisak et al., 2013*). There, T4 and T5 cells provide direct excitatory cholinergic input onto the dendrites of wide-field, motion-sensitive tangential cells as well as onto glutamatergic lobula plate interneurons that inhibit wide-field tangential cells in the adjacent layer (*Mauss et al., 2014*; *Mauss et al., 2015*). Through this circuit arrangement, lobula plate tangential cells depolarize to motion in their preferred direction (PD) and hyperpolarize in response to motion in the opposite or null direction (ND) (*Joesch et al., 2008*; *Schnell et al., 2010*). With T4 and T5 cells blocked, tangential cells lose all their direction selectivity (*Schnell et al., 2012*) and flies become completely motion-blind (*Bahl et al., 2013*; *Schilling and Borst, 2015*). This suggests that T4 and T5 cells are the elementary motion detectors and carry all directional information in the fly brain. Electrophysiological (*Behnia et al., 2014*), optical voltage (*Yang et al., 2016*) and Calcium recordings (*Meier et al., 2014*; *Serbe et al., 2016*; *Arenz et al., 2017*; *Strother et al., 2014*; *Strother et al., 2017*) from presynaptic medulla neurons revealed that none of them is directionally selective. Therefore, T4 and T5 cells are the first neurons in the visual processing chain that respond to visual motion in a direction selective manner.

Previous studies analyzed the mechanism underlying direction selectivity in T4 and T5 cells, yet arrived at different conclusions. Using apparent motion stimuli, one study found preferred direction enhancement to account for directional responses in T4 cells (*Fisher et al., 2015*). For T5 cells, the authors reported both enhancement for preferred and suppression for null direction sequences, but attributed the latter to circuit adaptation and not to the mechanism generating direction selectivity. The authors concluded that the dominant interaction producing direction selective responses in both T4 and T5 cells is a nonlinear signal amplification (*Fisher et al., 2015*). This conflicts with another report where spatio-temporal receptive fields of T5 cells were measured using white noise stimulation and reverse correlation. Based on ON and OFF subfields tilted in the space-time plane, T5 cells were concluded to incorporate both preferred direction enhancement and null direction suppression (*Leong et al., 2016*). This interpretation, however, suffers from a possible confusion of ON and OFF receptive subfields of T5 input neurons with the mechanism generating direction selectivity within T5 cells themselves. Addressing the same question, we recently applied apparent motion stimuli to one class of T4 cells that have upward as their preferred direction and, thus, project to layer 3 of the lobula plate. Using a telescopic stimulation technique to place the stimulus precisely onto the hexagonal lattice of the fly's eye (*Kirschfeld, 1967*; *Braitenberg, 1967*; *Franceschini, 1975*; *Schuling et al., 1989*), layer 3 T4 cells turned out to implement both mechanisms within different

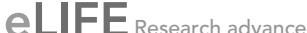

**Figure 1.** Receptive fields and responses to apparent motion stimuli of T5-cells. (a) The Hassenstein-Reichardt model incorporates PD enhancement only, realized by a multiplication. Left: Responses to individual light pulses ('Flicker') delivered at the two different positions. The responses are shifted according to the stimulus sequence used for the subsequent apparent motion stimuli. Middle: Responses of the model to apparent motion stimuli in preferred (upper row) and null direction (lower row, thick line = measured response, thin line = linear expectation, that is sum of responses to the single

*Figure 1 continued on next page*

*Figure 1 continued*

light pulses). Right: Nonlinear response component defined as the difference between measured response and linear expectation. (**b**) same as (**a**) but for a Barlow-Levick model. This model incorporates ND-suppression only, realized by a division. (**c**) Average responses of five T5 cells to flicker stimuli (stimulus size: 5 degree) delivered to different optical columns. In order to average the responses of different flies, the response patterns were aligned and normalized with respect to the maximum response (central column) and shown in a false color code. (**d**) Same as (**c**) but for T4-cells. Data represent the mean of 10 T4-cells from 10 flies (from *Haag et al., 2016*). (**e**) Responses of T5 cells to stimuli presented to the central column and simultaneously to one of the columns of the two surrounding rings. As in c, the responses of different flies were aligned with respect to the column eliciting the maximum response when stimulated individually and normalized to it. Depending on the location, simultaneous stimulation of a second column led to either a suppressed (blue colors) or an enhanced (red colors) response compared to the exclusive stimulation of the central column. The suppression is stronger on the null side of the T5 cells. Data represent the mean of 6 T5 cells from 6 different flies. (**f**) Same as e) but for T4-cells. Data represent the mean of T4-cells from 8 flies (from *Haag et al., 2016*).

DOI: https://doi.org/10.7554/eLife.29044.002

The following figure supplements are available for figure 1:

**Figure supplement 1.**

DOI: https://doi.org/10.7554/eLife.29044.003

**Figure supplement 2.** Responses of T4-cells and T5 cells to stimuli presented to the central column and simultaneously to one of the columns of the two surrounding rings.

DOI: https://doi.org/10.7554/eLife.29044.004

parts of their receptive field (*Haag et al., 2016*): While preferred direction enhancement was found to be dominant within the ventral part of the receptive field, a null direction suppression was significant in the dorsal part of their receptive field (*Haag et al., 2016*).

To resolve the conflicting evidences mentioned above and to test whether T5 cells are using the same or a different mechanism to compute the direction of motion as do T4 cells, we used the same strategy as in our previous account (*Haag et al., 2016*) and applied it to investigate the mechanism underlying direction selective responses of both T4 and T5 cells of all four directional tuning subtypes.

## Results

In a first set of experiments, we used the same driver line as in our previous study (*Haag et al., 2016*) expressing the Calcium indicator GCaMP6m (*Chen et al., 2013*) in both T4 and T5 cells projecting to layer 3 of the lobula plate and, hence, having upward motion as their preferred direction. Since T5 cells are known to be OFF sensitive, we used dark spots on a bright background projected onto the raster of optical columns via a telescope to stimulate the cells and recorded the fluorescence changes in the lobula plate. We started by measuring the flicker responses of T5 cells to optical stimulation of 19 individual columns, forming two rings surrounding a central column. In *Figure 1c*, the responses of five T5 cells were averaged and are shown in false color code overlaid on the columnar raster. T5 cells responded maximally to the stimulation of the central column, with about 50–100% amplitude to stimulation of the surrounding columns and about 20–50% to the next outer ring. An individual example trace and statistical evaluation of the responses are shown in *Figure 1—figure supplement 1*. Compared to T4 cells (*Figure 1d*, data replotted from *Haag et al., 2016*), the receptive field of T5 cells turned out to be broader with a somewhat stronger sensitivity within the surrounding columns. To explore spatial interactions within the receptive field of T5-cells, we stimulated the central column, simultaneously with one of surrounding columns. The results (*Figure 1e*) indicate a strong suppression of the response in the dorsal part of the receptive field compared to when the central column was stimulated alone, similar to what was found previously for T4 cells (*Figure 1f*, data replotted from *Haag et al., 2016*).

Experiments performed on layer 3 T4 cells with apparent motion stimuli revealed different mechanisms of direction selectivity in different parts of the receptive field (*Haag et al., 2016*): Two-pulse apparent motion stimuli in the dorsal part of the receptive field led to a null direction suppression, apparent motion stimuli in the ventral part evoked preferred direction enhancement. We asked whether we could find this spatial arrangement of null direction suppression and preferred direction enhancement in T5 cells as well. In order to measure that, we presented OFF stimuli to four neighboring columns along the dorsal-ventral axis (*Figure 2a,b*). The columns were chosen in relation to

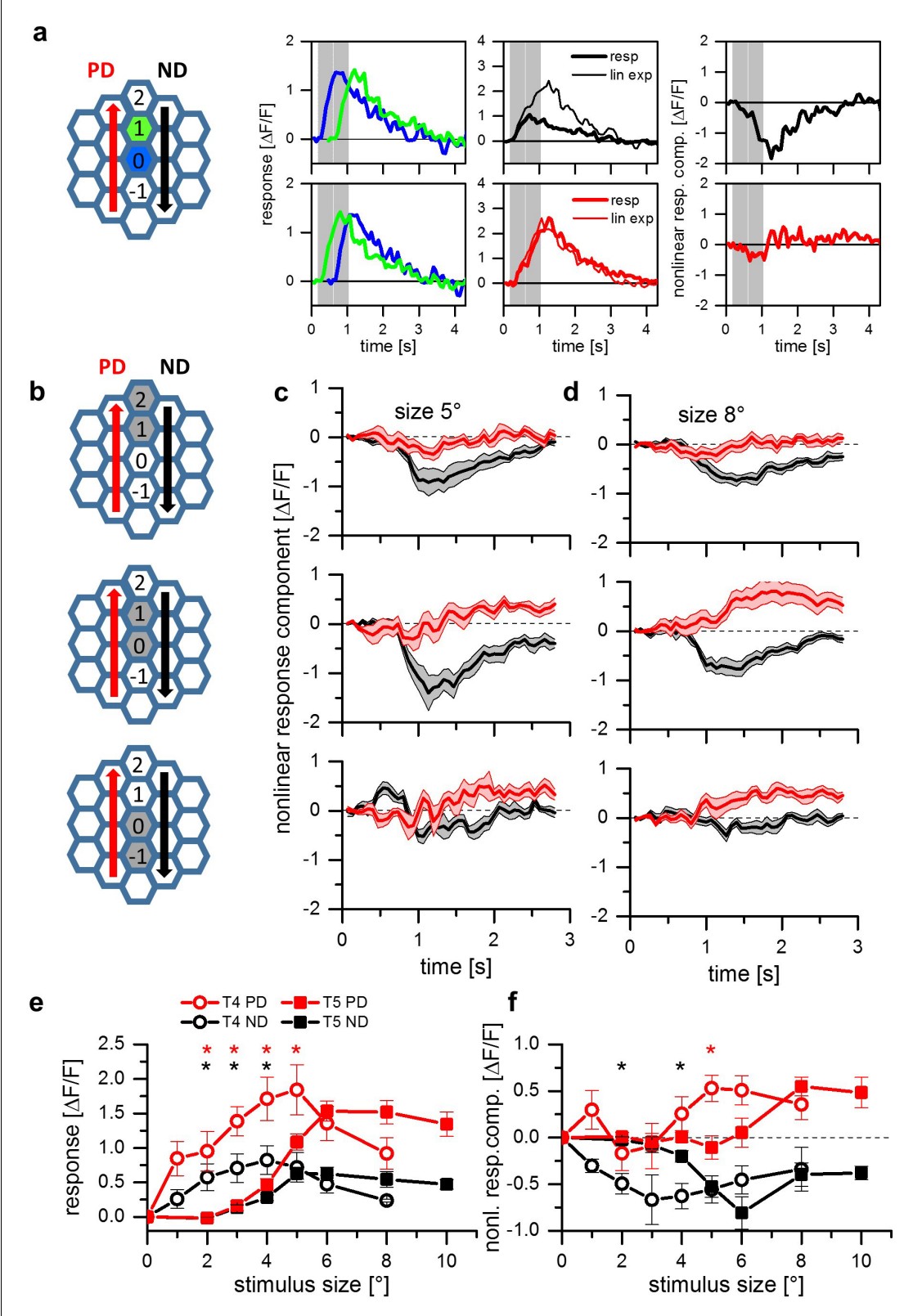

**Figure 2.** Apparent motion stimuli between adjacent cartridges. (**a**) Response of a single T5 cell recorded in a single sweep to two-step apparent motion stimuli. The schematic to the left shows the position of the two stimuli (blue and green shading). Left: Responses to individual light pulses ('Flicker') delivered at the two different positions. The responses are shifted in time according to the stimulus sequence used for the subsequent apparent motion stimuli.Middle: Responses of T5 to apparent motion stimuli in preferred and null direction (thick line = measured response, thin
*Figure 2 continued on next page*

*Figure 2 continued*

line = linear expectation, that is sum of responses to the single light pulses). Right: Nonlinear response component defined as the difference between measured sequence response and linear expectation. The responses are the mean obtained from n = 3 stimulus repetitions. (**b**) Two-step apparent motion stimuli were shown at three different position in the receptive field of T5 cells. The stimulus consisted of light off pulses positioned on one column for 472 ms, immediately followed by a light off pulse for 472 ms to the upper, neighboring cartridge. The same stimuli were repeated along the opposite direction. (**c**) Nonlinear response component, that is the difference between sequence response and the sum of the responses to the individual pulses, as a function of time for a stimulus size of 5 degree. Apparent motion stimuli delivered to the upper two cartridges resulted in a null direction suppression and no preferred direction enhancement. Apparent motion stimuli in the lower cartridges did not lead to a deviation from the linear expectation. For all three stimuli no preferred direction enhancement could be found. Data represent the mean ± SEM in 6 T5-cells measured in 6 different flies. (**d**) Same as d, but with a stimulus size of 8 degree. In contrast to the results for a smaller stimulus size, we found preferred direction enhancement for stimulation of the lower and the central pair of columns. Data shows the mean ± SEM in 10 T5-cells measured in 8 different flies. (**e**) Responses of T4 (open circles) and T5 cells (closed squares) to apparent motion stimuli in preferred (red colors) and null direction (black colors) between the two central columns 0 and 1. Compared to the responses of T4, T5 responses to two-pulse sequences along the preferred (PD) and the null (ND) direction are shifted to larger stimulus sizes. Data represent the mean values ± SEM of 10 T5 cells measured in 5 flies and of 7 T4 cells in 4 different flies, respectively. Black asterisks represent statistically significant (t-test, p-value<0.05) differences for null-direction responses of T4-cells and T5-cells, red asterisks for preferred direction responses. (**f**) Nonlinear response components of T4 and T5 cells. Same dataset as in *Figure 1e*.

DOI: https://doi.org/10.7554/eLife.29044.005

the column that elicited the strongest response in each cell (central column 0). We then tested T5 cells with two consecutive light OFF pulses of 472 ms duration in immediate succession. Each light pulse was positioned on one of two neighboring columns resulting in three stimulus sequences (*Figure 2b*). To extract the nonlinear response component, we subtracted the sum of the responses to the individual stimuli from the response to the apparent motion sequence. Example traces from an individual experiment stimulating columns 0 and 1 are shown in *Figure 2a*. Using the same spot diameter as in our previous account, that is 5 degree, we found null direction suppression for stimulation of the central (column 0 and 1) and the dorsal (column 1 and 2) pairs, and only a slight, if any, sign of preferred direction enhancement for stimulation of the ventral pair (column −1 and 0) (*Figure 2c*). This changed when we enlarged the spot size from 5 to 8 degree. Now, in addition to null direction suppression for the central and dorsal stimulus pairs, preferred direction enhancement for the ventral and central stimulus pairs was observed (*Figure 2d*). This result mirrors our previous finding for T4 cells where both null direction suppression and preferred direction enhancement was found to account for direction selectivity (*Haag et al., 2016*). In further agreement with T4 cells, these two mechanisms are spatially separated, with null direction suppression on the null and preferred direction enhancement on the preferred side of the receptive field (*Haag et al., 2016*).

The above experiments indicate a different dependence of null direction suppression and preferred direction enhancement on the diameter of the stimulus spot in T5 cells. To measure this dependence in a gradual way, we again used apparent motion stimuli and varied the size of the stimuli from 1 to 10 degree. Since stimuli centered on the central column pair (0 and 1) resulted in both types of nonlinearity (*Figure 2d*, middle graph), we presented apparent motion stimuli with different stimulus sizes to these central columns only. To compare the results of T5 cells with the ones of T4 cells, the stimulus set consisted of either bright pulses on a dark background (for T4 cells) or dark pulses on a bright background (for T5 cells). *Figure 2e* shows the responses of T4 (circle symbols) and T5 (square symbols) to apparent motion stimuli as a function of the stimulus size for preferred (PD, red traces) and null (ND, black traces) direction sequences. For both directions of motion, T4 cells respond to smaller stimuli than T5 cells. The strongest response in T4 can be found for stimulus sizes of 4 to 5 deg. For stimulus sizes beyond these values, the responses of T4 cells decline. In contrast, T5 cells only start responding at these stimulus sizes and plateau for larger values. When instead of the response the nonlinear response component is plotted as a function of the stimulus size (*Figure 2f*, same symbol and color code as in *Figure 2e*), both preferred direction enhancement and null direction suppression become apparent, with both curves shifted to larger stimulus sizes for T5 cells. Furthermore, for both cell types, null direction suppression peaks at smaller stimulus sizes than preferred direction enhancement.

The results presented so far point towards a common mechanism for T4 and T5 cells underlying direction selectivity. However, the experiments on T4 and T5 cells were confined to those that terminate in layer 3 of the lobula plate. In order to investigate whether the properties described above

generalize to T4 and T5 cells of all four tuning categories, we next used a fly line expressing GCaMP5 in T4 and T5 cells projecting to all four layers. Similar to the experiments shown in *Figure 2c and d*, two-step apparent motion stimuli consisting of ON-ON pulses of 5 degree diameter for T4 cells as well of OFF-OFF pulses of 8 degree diameter for T5 cells were presented in three adjacent pairs of columns aligned to the column that elicited the strongest flicker response (*Figure 3a*). As before, the stimulus protocol consisted in the presentation of individual stimulus pulses for the calculation of the linear expectation as well as in the presentation of the two-pulse sequences, to measure the sequence response. From the latter, the linear expectation was subtracted to obtain the nonlinear response component. The time traces of these nonlinear response components are shown in *Figure 3b and c*. For T4 (*Figure 3b*) and T5 (*Figure 3c*) cells projecting to all layers, we found both preferred direction and null direction suppression, with a spatial separation that follows the same pattern: on the preferred side of the receptive field, a clear preferred direction enhancement was observed without any null direction suppression (*Figure 3b and c*, left column). In the center of the receptive field, both preferred direction enhancement and null direction suppression prevailed (*Figure 3b and c*, center column). On the null side of the receptive field, only null direction suppression was detectable (*Figure 3b and c*, right column). To investigate possible differences between T4 and T5 cells and between cells with different directional tuning, we performed a 3-way ANOVA test. Choosing a significance level of p=0.05, no significant differences were found, neither between T4 and T5 cells, nor between the neurons projecting to the four different layers. In *Figure 3d* (T4 cells) and *3e* (T5 cells), the nonlinear response components are shown as averaged between 1 and 2 s of the time courses shown above, as well as averaged across the cells from all four layers. On these data, two-sided t-tests were performed between T4 and T5 cell responses for each individual stimulus condition. Choosing again a significance level of p=0.05, no differences were found between T4 and T5 cell responses for 5 out of 6 stimulus conditions. Only the response amplitude of T4 cells to null direction stimulus sequences from column 2 to 1 was found to be significantly smaller than the respective value of T5 cells.

The results from apparent motion experiments reported so far suggest a common directional tuning mechanism for T4 and T5 cells for all cardinal directions. One, thus, would expect identical high degrees of direction selectivity in response to moving gratings mechanism within the different layers of the lobula plate. To test this directly, we used the same fly line as above expressing in both T4 and T5 cells of all four layers and presented grating motion along all four cardinal directions on a screen. For each pixel, we first calculated the vector sum of the responses and represented the vector angle in false color. The resulting image from one example fly is shown in *Figure 4a*. Clearly, the preferred direction is extremely homogeneous with little variation within each layer. We repeated such experiments in five different flies and determined the distribution of all preferred directions obtained from the whole data set. The histogram (*Figure 4b*) reveals exactly four sharp peaks separated by 90 degrees, corresponding to the preferred directions of T4 and T5 cells within each of the four layers. This transition from the hexagonal coordinates of the fly eye to Cartesian coordinates is likely to occur on the dendrites of T4 and T5 cells by their sampling from appropriately grouped columns (*Takemura et al., 2017*). From the same data set, we calculated a direction selectivity index for each pixel within each layer as the difference of the responses to preferred and null direction, divided by the sum of the responses. We then determined the mean direction selectivity for each layer from each fly and averaged the resulting values across the different experiments. The results reveal an extremely high degree of direction selectivity of about 0.8 that is almost identical within each layer (*Figure 4c*). To measure direction selectivity separately for T4 and T5 cells, we stimulated the flies with ON and OFF edges instead of gratings and obtained similar values of about 0.8 on average (*Figure 4d*).

## Discussion

Having analyzed the mechanisms underlying direction selectivity of all fours subtypes of both T4 and T5 cells, we found a common scheme that pertains to all of these cells: regardless of the directional tuning and the contrast preference for ON or OFF stimuli, elementary motion-sensitive neurons in *Drosophila* implement a preferred direction enhancement on the preferred side and a null direction suppression of input signals on the null side of their receptive field.

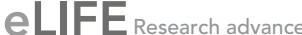

**Figure 3.** A common mechanism for direction selectivity in all four subtypes of T4 and T5 cells. (**a**) Pictograms indicating the stimulus positions and the preferred and null-direction of the respective layer. (**b**) Nonlinear response components of T4-cells to apparent motion stimuli in different layers of the lobula plate. For T4 cells projecting to all four layers, preferred direction enhancement and null direction suppression are found to be spatially distributed within the receptive field such that enhancement is found on the preferred side while suppression is predominant on the null side of the

*Figure 3 continued on next page*

*Figure 3 continued*

receptive field. Data represent the mean ± SEM of 6, 8, 7 and 9 T4 cells (from layer 1–4). (**c**) Nonlinear response components of T5 cells to apparent motion stimuli in different layers of the lobula plate. Data represent the mean ± SEM of 8, 5, 6 and 13 T5 cells (from layer 1–4).

DOI: https://doi.org/10.7554/eLife.29044.006

ON and OFF pathways seemed to have adapted to the asymmetry of luminance distributions found in the real world. Consequently, functional differences between ON and OFF pathways have been described in the mammalian retina and in flies as well (*Ratliff et al., 2010*; *Clark et al., 2014*; *Baden et al., 2016*; *Leonhardt et al., 2016*). In fly motion vision, our finding of a common mechanism for T4 and T5 cells suggests the above mentioned asymmetries to rely on quantitative instead of qualitative differences, such as different time-constants used by the ON and the OFF pathway (*Leonhardt et al., 2016*; *Arenz et al., 2017*). One difference between T4 and T5 cells found in this

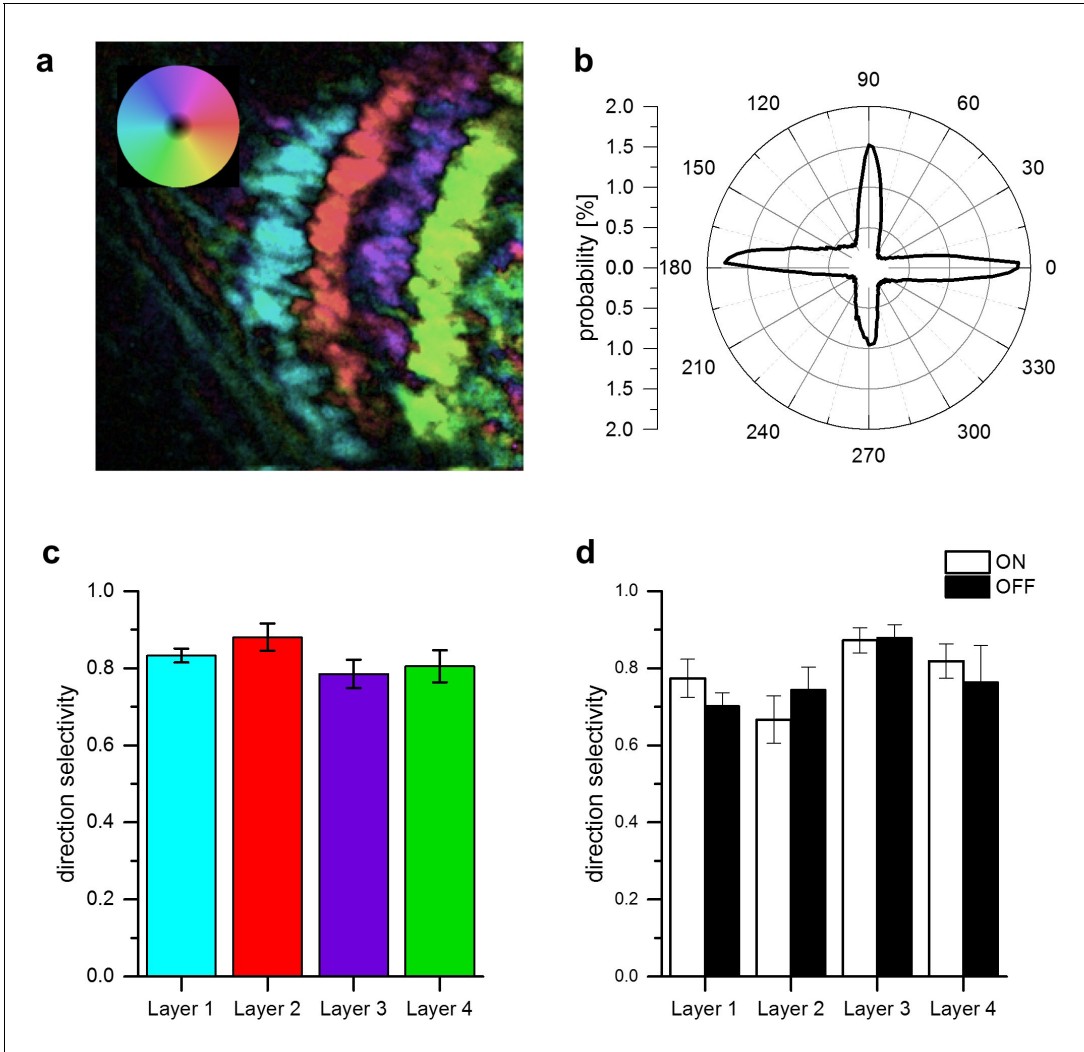

**Figure 4.** Directional tuning and selectivity of T4 and T5 cells. (**a**) Example of directional tuning to grating motion as determined by the vector sum of responses to grating motion along four cardinal directions. All neurons within each layer have almost identical preferred directions. (**b**) Histogram of preferred directions within all four layers. Clear peaks appear at the four cardinal directions. Data were obtained from 5 different flies. (**c**) Direction selectivity within each layer, as defined by the difference between the preferred and null direction responses, divided by the sum. Data represent the mean ± SEM obtained from 5 flies (same data set as in b). (**d**) Same as c, but flies were stimulated by ON and OFF edges, respectively. Data represent the mean ± SEM obtained from 3 flies.

DOI: https://doi.org/10.7554/eLife.29044.007

study relates to the dependence of the directional motion signal on the spot size in the apparent motion paradigm (*Figure 2e,f*). Given a half-width of the photoreceptor acceptance angle of approximately 5 degree in *Drosophila* (*Götz, 1965*), any stimulus is spatially low-pass filtered by a Gaussian with 5 degree full width at half maximum. Accordingly, enlarging the spot size will have two different effects: first, it will lead to an increasing peak intensity at the column where the stimulus spot is centered on, and second, it will lead to an increasing activation of neurons in neighboring columns. Which of these two effects is responsible for the higher threshold of T5 cells compared to T4 cells, and whether the sensitivity difference is in the input neurons or in T4/T5 cells themselves, cannot be decided by the present study.

In any case, our finding readily explains the high degree of direction selectivity found already at the processing stage where direction-selective signals first arise: neither a signal enhancement for preferred direction sequences nor a signal suppression for null direction sequences by itself would lead to such a strong direction selectivity as observed experimentally with large signals for preferred direction motion and zero responses for null direction motion (*Maisak et al., 2013*; *Fisher et al., 2015*). In analogy to the results presented for layer 3 T4 cells (Figure 5 in *Haag et al., 2016*), the responses of all T4 and T5 cells can be captured in algorithmic terms by a common mechanism, using a delayed, low-pass filtered input on the preferred side enhancing a fast, central input, with the result being suppressed by again a low-pass filtered input on the null side.

At the next processing stage, that is at the level of lobula plate tangential cells, the signals of oppositely tuned T4 and T5 cells become subtracted via inhibitory lobula plate interneurons (*Mauss et al., 2015*). This process, in a way, replicates the action of null direction suppression implemented on the dendrites of T4 and T5 cells. Since both mechanisms, that is the combination of preferred direction enhancement and null direction suppression on the dendrites of T4 and T5 cells as well as the subtraction of oppositely tuned T4 and T5 cells on the dendrites of tangential cells lead to high degree of direction selectivity at the output of the system, one might ask about the functional advantage of such a dual strategy. This question can be answered by either blocking null direction suppression on the T4/T5 cells dendrite or blocking the inhibitory lobula plate interneurons. The latter experiment has indeed been done, and the results revealed a loss of flow-field specificity of the tangential cells, due to the lack of inhibition caused by the non-matching part of the optic flow field (*Mauss et al., 2015*). For the converse situation, no experimental data exist so far and one has to rely on computer simulations (*Haag et al., 2016*, Figure 5). They suggest that a high direction selectivity is retained in tangential cells. This high degree of direction selectivity, however, rests on the relatively small differences between large, but poorly tuned signals and, thus, would be highly prone to noise. Improving the direction-tuning already at the level of T4 and T5 cells by the additional null-direction suppression should, therefore, increase the system's robustness to noise.

Our results open the door to the next level question about its neural implementation. Here, a recent connectomic study identified the major interneurons providing synaptic input to T4 cells as well as their placement on the dendrite (*Takemura et al., 2017*). Takemura and colleagues describe 4 columnar cell types, that is Mi1, Mi4, Mi9 and Tm3 as the major input elements to T4 cells, in addition to columnar cell types C3, TmY15, a wide-field neuron CT1 and other T4 cells with identical preferred direction. Columnar input neurons contact the T4 cell dendrite in a way that depends on the direction tuning subtype: while Mi9 synapses are clustered on the preferred side of the dendrite, Mi1 and Tm3 synapse on the central part and Mi4 are found predominantly on the null side. Most interestingly, the dynamic response properties of these different types of T4 input neurons match their position on the dendrite to suggest a specific function in the detector model discussed above: Mi9 and Mi4 indeed exhibit the temporal low-pass properties postulated for the inputs on the preferred and the null side, while Mi1 and Tm3 display fast band-pass properties needed for the central input (*Arenz et al., 2017*). This proposed correspondence needs to be tested by blocking individual input cell types and measuring the resulting effect on direction selectivity in T4 cells. Specifically, one would expect to abolish preferred direction enhancement when blocking Mi9, while blocking Mi4 should lead to a loss of null direction suppression. It is, however, important to stress that the effect of such blocking experiments is expected to be quite specific and directly observable only in directional responses of T4 cells: due to further network processing involving a subtraction of T4 cell signals with opposite directional tuning at the level of tangential cells (see previous paragraph, and *Mauss et al., 2015*), the effect might be far more subtle when downstream cells or behavior are used as a read-out (*Strother et al., 2017*). Nevertheless, blocking the central inputs Mi1 and Tm3

while recording from tangential cells revealed that Mi1 cells are absolutely essential for proper functioning of the ON pathway under all stimulus conditions tested, while blocking Tm3 only led to a loss of sensitivity for high edge velocities (*Ammer et al., 2015*). With respect to the polarity of the synapses of the various T4 input neurons, the correspondence outlined above predicts that Mi1 and Tm3 are excitatory while Mi4 should be inhibitory. In line with this, recent studies suggest a cholinergic phenotype in Mi1 and Tm3 (*Pankova and Borst, 2017*; *Takemura et al., 2017*) and a GABAergic one in Mi4 (*Takemura et al., 2017*). Seemingly in contrast to an enhancing action of Mi9 postulated above, this cell was found to be OFF sensitive (*Arenz et al., 2017*). However, Mi9 turned out to be immune-positive for the vesicular Glutamate reporter VGlut (*Takemura et al., 2017*). Together with the inhibitory action of Glutamate via the GluCl channel, well documented for other neurons of the Drosophila CNS (*Liu and Wilson, 2013*; *Mauss et al., 2014*; *Mauss et al., 2015*), this raises the possibility that Mi9 enhances the input from Mi1 and Tm3 onto T4 by a release from inhibition.

As for T4 cells, the major input neurons to T5 cells were identified by an EM study (*Shinomiya et al., 2014*). There, trans-medulla neurons Tm1, Tm2, Tm4, and Tm9 were found to make up for about 80% of all input synapses to T5 cell dendrites. However, the exact placement of the different inputs on the dendrite and, hence, the relative position of their receptive fields could not be determined by this report. As for their dynamic properties, only one of the cell types, Tm9, reveals low-pass characteristics, while all others (Tm1, Tm2 and Tm4) can be described as band-pass filters with different time-constants (*Meier et al., 2014*; *Serbe et al., 2016*; *Arenz et al., 2017*). In analogy of the arrangement of input neurons of T4 cells, *Arenz et al., 2017* found that placing the two slowest cells (Tm1 and Tm9) on the outer arms and the fast Tm2 cell on the central arm of the three-input detector gives rise to a motion detector that fits real T5 cells both with respect to their high degree of direction selectivity and their temporal tuning optimum. Since, in contrast to T4 cells, the position of these interneurons on the dendrite of T5 cells is less well known so far, no prediction can be made whether blocking of Tm1 should lead to a loss of preferred direction enhancement and blocking of Tm9 to a loss of null direction suppression, or the other way round. Therefore, as is the case with T4 cells, further experiments are needed to determine which cell is playing which role in the functional context of preferred direction enhancement and null direction inhibition determined by the present study.

In summary, thus, we have found a common, uniform mechanism of direction selectivity for T4 and T5 cells that consists of combination of preferred direction enhancement and null direction suppression in different location of their receptive field, precisely related to their directional tuning. Mapping the different input neurons to T4 and T5 cells to their specific function in this context represents the next step of the analysis. The major challenge for future experiments will then consist in understanding the biophysical mechanisms underlying enhancement and suppression. Here, different ideas have been discussed in the past (*Torre and Poggio, 1978*; *Koch and Poggio, 1992*; *Gabbiani et al., 2002*), and the different thresholds for preferred direction enhancement and null direction suppression described above might be an important result to decide between the various possibilities. These can now be tested at the molecular level using genome editing techniques available in Drosophila (*Venken et al., 2011*; *Zhang et al., 2014*; *Fisher et al., 2017*; *Pankova and Borst, 2017*).

## Materials and methods

### Flies

(*Drosophila melanogaster*) were raised at 25°C and 60% humidity on a 12 hr light/12 hr dark cycle on standard cornmeal agar medium. For calcium imaging of T5 cells, flies were used expressing the genetically-encoded calcium indicator GCaMP6m (*Chen et al., 2013*) in T4/T5 neurons with axon terminals predominantly in layer 3 of the lobula plate (w⁻; Sp/cyo; VT50384-lexA, lexAop-GCaMP6m/TM6b). For the imaging experiments of T4 and T5 cells in the four layers of the lobula plate we used flies expressing the calcium indicator GCaMP5 in both T4 and T5 cells in all layers of the lobula plate (w⁻; +/+; UAS-GCaMP5, R42F06-GAL4/UAS-GCaMP5,R42F06-GAL4).

## Calcium imaging

Fly surgery was performed, and the neuronal activity was measured from the left optical lobe on a custom-built 2-photon microscope (*Denk et al., 1990*) as previously described (*Haag et al., 2016*). Images were acquired at a resolution of 64 × 64 pixels and at a frame rate of 15 Hz with the Scan-Image software (*Pologruto et al., 2003*) in Matlab.

## Optical stimulation

Stimulation with a telescopic stimulus device was similar to that used in our previous study (*Haag et al., 2016*). For the experiments shown in *Figure 4*, a regular stimulus display was used as described in (*Arenz et al., 2017*). The gratings had a spatial wavelength of 30 deg, a contrast of 100%, a mean luminance of 34 cd/m$^2$ and was moving along one of the four cardinal directions at 30 deg/sec.

## Experimental protocol

In order to discriminate between T4 and T5 cells we stimulated single optical columns with bright pulses on a dark background. The cells were selected based on their response to light-on stimuli. While T4 cells respond to the onset of a light pulse, the T5 cells respond to the light-off. For the experiments the stimuli consisted either of dark pulses on a bright background (T5 cells) or bright pulses on a dark background (T4 cells). The pulses had a duration of 472 ms. At each position, three stimulus presentations were delivered. The resulting responses were averaged and the peak of the averaged response was taken. Apparent motion stimuli consisted of consecutive light stimuli to two neighboring cartridges. The second stimulus was presented right after the first turned off, resulting in a delay from onset to onset of 472 ms.

## Data analysis

was performed offline using custom-written routines in Matlab. Regions of interests (ROIs) were selected by hand of the lobula plate. Time courses of relative fluorescence changes (ΔF/F) were calculated from the raw imaging sequence. Responses to the stimulus were baseline-subtracted, averaged across repetitions, and quantified as the peak responses over the stimulus epochs. Those responses were averaged across experiments. Where indicated, responses were normalized to the maximum average response before averaging. For the apparent motion experiments, non-linear response components were calculated as the differences of the time-courses of the responses to the apparent motion stimuli and the sum of the appropriately time-shifted responses to flicker stimuli at the corresponding positions.

## Acknowledgements

We thank Georg Ammer, Alexander Arenz and Alex Mauss for critically reading the ms. This work was supported by the Max-Planck-Society and the Deutsche Forschungsgemeinschaft (SFB 870).

## Additional information

### Competing interests

Alexander Borst: Reviewing editor, *eLife*. The other authors declare that no competing interests exist.

### Funding

| Funder | Grant reference number | Author |
| --- | --- | --- |
| Max-Planck-Gesellschaft | | Juergen Haag Abhishek Mishra Alexander Borst |
| Deutsche Forschungsgemeinschaft | SFB 870 | Juergen Haag Abhishek Mishra Alexander Borst |

The funders had no role in study design, data collection and interpretation, or the decision to submit the work for publication.

## Author contributions
Juergen Haag, Conceptualization, Data curation, Software, Investigation, Visualization, Writing—review and editing; Abhishek Mishra, Investigation; Alexander Borst, Conceptualization, Funding acquisition, Writing—original draft, Project administration, Writing—review and editing

## Author ORCIDs
Juergen Haag https://orcid.org/0000-0002-6535-0103
Abhishek Mishra http://orcid.org/0000-0002-1933-1251
Alexander Borst http://orcid.org/0000-0001-5537-8973

## Decision letter and Author response
Decision letter https://doi.org/10.7554/eLife.29044.010
Author response https://doi.org/10.7554/eLife.29044.011

## Additional files
### Supplementary files
• Transparent reporting form
DOI: https://doi.org/10.7554/eLife.29044.008

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
