## [Decision Letter]

Thank you for submitting your article "A Common Directional Tuning Mechanism of *Drosophila* Motion-Sensing Neurons in the ON and in the OFF Pathway" for consideration by *eLife*. Your article has been favorably evaluated by David Van Essen (Senior Editor) and three reviewers, one of whom is a member of our Board of Reviewing Editors. The reviewers have opted to remain anonymous.

The reviewers have discussed the reviews with one another and the Reviewing Editor has drafted this decision to help you prepare a revised submission.

This is a follow-up on a paper published in *eLife* last year probing the circuitry for motion detection in the fly visual system. All of the reviewers agreed that the paper was a good candidate for a research advance: it does not break new ground, but shows that the results of the previous paper extend to multiple types of T4 cells and to T5 cells. The reviewers also agreed that the current paper could be considerably stronger. The following are the main points emphasized in the discussion among the reviewers. More details, and additional issues, can be found in the individual reviews.

1) The paper could be much stronger if it was extended to explore some of the issues that are highlighted by the finding that the motion detection mechanisms are shared across motion sensor types. For example, what is the importance of the asymmetries between On and Off motion sensors? And what is the impact of having two mechanisms shaping directional selectivity rather than one? Another extension that would enhance the paper would be to explore the temporal and spatial dependence of each mechanism in more detail.

2) The Introduction needs revisions. At present, the Introduction quickly delves into details of the motion circuitry, without a clear summary of the evidence for and against the models tested. A broader Introduction would help a general audience appreciate the paper. In addition, other similar work needs to be presented in a more objective manner, before summarizing the results of the paper that this follows up on.

3) The figures could be improved by including more labels and by providing a better connection to underlying measurements. At present some of the results are too derived and the connection to the measured data is unclear. The comparison with the previous T4 results could also be improved by showing some of the T4 and T5 data side by side.

4) The Discussion could be improved by clarifying the key testable predictions made from the present work.

*Reviewer #1:*

This is a follow up on a paper from a year ago (Haag et al.) describing the presence of both preferred and null direction mechanisms for direction sensitivity in one type of fly T4 cell. The present paper extends this analysis, using an identical approach, to the other T4 cells and to T5 cells. I am not expert in fly vision, and thus cannot place this work in the context of the literature. The paper does not provide fundamentally new results, but shows that the results from Haag et al. apply more broadly to the T4 and T5 cells. The presentation of the work can be improved in several ways:

1) Introduction

The Introduction is not well suited for a broad audience. It starts with a long paragraph that almost immediately jumps into the anatomy of the fly motion circuits, but does not provide enough detail about the anatomy for someone unfamiliar with the organization of the fly visual system. It is, for example, hard to generate a clear picture from the introduction about how the circuitry for T4 and T5 cells differs. I think the Barlow-Levick and Hassenstein-Reichardt models, and the lack of direct tests of them, could be introduced earlier to make the central question of the paper clearer. Then maybe the anatomical information could be in a separate paragraph, once it has been clarified that you are interested in both excitatory and inhibitory circuits. Related to above, the second paragraph is clearly critical to the paper. I would consider adding a schematic to Figure 1 that shows the two models and the logic of the paired flash tests. I think that would go a long way to setting the paper up for a general reader.

2) Comparison with past work.

The third paragraph of the Introduction describes the diversity of findings about how direction is computed. I think this could be done in a more objective manner. At present, the paragraph first presents the results and logic of the Haag et al. paper in some detail, and then presents findings of other work only in this context. Further, the language used to describe the other work is not objective (e.g. "but blamed the latter.…"). I think a better way to present this would be to say that there are discrepancies in existing studies, describe those different findings, and then describe the Haag et al. results. This would naturally set up the present work at the end of the paragraph.

3) Figure 1.

Figure 1 do not convey the nature of the underlying data and do not provide a sense of the reliability of the findings. First, some examples of the raw traces for a few selected locations would be helpful. Second, making the color schemes comparable between A and B is important since you are asking the reader to compare the two. Third, some quantification of the results with error bars is needed (the first paragraph of the Results section compares T4 and T5 but very qualitatively). Fourth, showing at least one example of the traces to each spot separately and together prior to subtraction would help a reader appreciate both Figure 1 and the differences between those and 1F. Fifth, some statistical comparison is needed for the comparisons in Figure 1(see Results, third paragraph).

4) Single vs. multiple columns.

The larger spots should activate multiple columns. Is this essential for T5 activation? This could get mentioned when it first comes up and the text could more clearly state that. For example, you say "from 1 to 10 degrees. Since stimuli delivered to the central column[…]" I would say "stimuli centered on the central column" since they are not restricted to one column. Generally this issue could be described and discussed more clearly.

*Reviewer #2:*

The origins of direction selectivity in the visual system has been the subject of study across systems. In *Drosophila melanogaster*, T4 and T5 cells of the optic lobe show the first directionally selective responses. In a 2016 study, Haag et al. provided evidence suggesting that preferred direction enhancement and null-direction suppression, two historically competing model mechanisms of producing direction selective neural responses, both contribute to direction-selectivity in T4 cells of the *Drosophila* optic lobe. Here, the authors rely on similar methods and analyses to extend their findings to T4 cells selective for all preferred directions, and, importantly, to T5 cells of the OFF pathway.

As with the previous study, the authors use calcium imaging as the basis of their claims, and use a similar procedure to describe null-direction suppression: normalization relative to control responses that are obtained by stimulating the central column of a T5 RF. Stimulating elsewhere with or without a delay reveals a reduction/increase relative to this max response, which is used to evaluate preferred direction (PD) enhancement/null direction (ND) suppression. These experiments find that T4 cells corresponding to all four PDs reveal similar PD enhancement and ND suppression (matching Haag et al., 2016). Additionally, although T5 cells show some differences relative to T4 (more on that later), the results from these cells are in broad agreement that the same combination of PD enhancement and ND suppression. The study is a logical extension of the previous work and certainly deserves to be published. I have only a few comments.

1) This study often refers to specific results and figure panels from the 2016 study, but comparisons would be made significantly easier if the results were presented side-by-side with the same color maps/scaling, whether this is done as part of supplementary figures or by repeating previous Figure panels in the new color maps.

2) The authors discuss the differences in stimulus-size dependence between T4 and T5 cell responses (Figure 1). However, Figure 1 seems to show a significant difference between the stimulus-size dependence of ND suppression and PD enhancement within T5 cells. How do the authors interpret this?

3) In Discussion, the authors discuss at length the issue of neural implementation and mechanistic underpinnings for their model/results. For some part of this discussion, they refer to EM reconstructions of different medulla inputs to T4/T5 cells to potentially underlie their findings of PD enhancement on the null side of T4/T5 RFs and ND suppression on the preferred side of the RFs. With T5 cells in particular, they propose specific layouts of inputs and delays (and filtering) that would be consistent with their findings. Given that their study relies on calcium imaging and normalized responses to infer ND suppression, however, there is still some room for uncertainty about interpretation and mechanism. Thus, it would be helpful to readers to know what key properties of inputs they would consider inconsistent with the proposed spatially segregated combination of HR and BL type mechanism that their evidence suggests underlies T4/T5 responses (particularly T5, since there is some uncertainty about spatial location of inputs in this case, even from EM). Essentially, what would falsify their current model of direction selectivity? For example, does the model rely on some of the synaptic inputs to be inhibitory to produce ND suppression, or is that not a constraint? Essentially, a more in-depth discussion of caveats and laying out what key experiments would test these models may help better ground some of the interesting and intense debates in the field.

*Reviewer #3:*

The authors test two models of direction selectivity in the fly visual system: the Barlow-Levick model, where inhibition is increased during null direction motion, and the Hassenstein-Reichardt model, where excitation is increased during preferred direction motion. They employ an apparent motion stimulus, which allows them to discriminate the two models in different regions of the receptive field of T4 and T5 cells. They find that preferred direction enhancement (HR model) dominates in the null side, while null-inhibition (BL model) in the preferred-side of the DS neurons receptive field.

Overall, the experiments are carefully done and the results seem fairly unambiguous. However, given that such a circuit organization was described by this group last year for the T4 cells (Haag et al., *eLife*, 2016), I'm having a hard time appreciating the major conceptual advancement to be made from the current work.

Demonstrating similar circuit mechanism for DS exists for the OFF system is undoubtedly an important step, but in and of itself it might appear as a detail for the general reader. The relevance of having two mechanisms for DS in different parts of the RF, and/or how the more subtle differences observed between ON and OFF pathways (e.g. Figure 1: size) work to improve DS coding for more complex stimuli (with dark/ and light), are important questions that have not been addressed.

Other comments:

Recently, orientation selective inhibitory field was described to sharpen DS in fly neurons by the Clandinin's group. It would be useful to discuss the reasons for the different results/model presented here.

It is also important to discuss the anatomical arrangement of the Mi4 and Mi9 cells that are known to inhibit T4s (i.e. are they displaced to the preferred side?)

The nomenclature is confusing. Usually, the preferred side refers to the side at which the preferred stimulus enters a DS neurons receptive field. The preferred and null sides have not been defined and appear to be opposite to the normal convention.

Figure 4. The DS tuning appears to be extremely sharp which makes it challenging to understand how 'in between' directions (e.g. 60,120, 210, 315 degrees) would be computed.

The nonlinear components for increased null direction inhibition or increased preferred direction excitation appear to have different time courses. In many cases (especially in Figure 1, but maybe it's less stereotyped in Figure 2–Figure 3), the increases in ND inhibition occur earlier and decay faster than the increases in PD excitation. Were any experiments done investigating the relative time course of the two mechanisms? It would be useful to discuss some possible reasons for why there are temporal differences between the two mechanisms.

[Editors' note: further revisions were requested prior to acceptance, as described below.]

Thank you for submitting your article "A Common Directional Tuning Mechanism of *Drosophila* Motion-Sensing Neurons in the ON and in the OFF Pathway" for consideration by *eLife*. Your article has been reviewed by a Reviewing Editor and David Van Essen as the Senior Editor.

The revisions did a good job addressing most of the comments raised in review. One substantive issue remains: a need for statistical analyses to accompany the conclusions reached from Figure 1 and 2. This should particularly include the comparisons of T4 and T5 cells.

---

## [Author Response]

*This is a follow-up on a paper published in eLife last year probing the circuitry for motion detection in the fly visual system. All of the reviewers agreed that the paper was a good candidate for a research advance: it does not break new ground, but shows that the results of the previous paper extend to multiple types of T4 cells and to T5 cells. The reviewers also agreed that the current paper could be considerably stronger. The following are the main points emphasized in the discussion among the reviewers. More details, and additional issues, can be found in the individual reviews.*
*1) The paper could be much stronger if it was extended to explore some of the issues that are highlighted by the finding that the motion detection mechanisms are shared across motion sensor types. For example, what is the importance of the asymmetries between On and Off motion sensors? And what is the impact of having two mechanisms shaping directional selectivity rather than one? Another extension that would enhance the paper would be to explore the temporal and spatial dependence of each mechanism in more detail.*

We have extended the discussion along all these lines, in particular with respect to the question about the advantage of having two mechanisms sharpening direction selectivity over just one. We now write: ‘At the next processing stage, i.e. at the level of lobula plate tangential cells, the signals of oppositely tuned T4 and T5 cells become subtracted via inhibitory lobula plate interneurons (Mauss et al., 2015). […] As another advantage, increasing the directional specificity of T4 and T5 cells might in addition be beneficial with respect to energy consumption of the cells.’

We have started doing experiments that address the temporal and spatial domain of each mechanism, together with blocking inputs to T4 and T5. This, however, is taking quite some time before being finished and will represent the content of a future publication.

*2) The Introduction needs revisions. At present, the Introduction quickly delves into details of the motion circuitry, without a clear summary of the evidence for and against the models tested. A broader Introduction would help a general audience appreciate the paper. In addition, other similar work needs to be presented in a more objective manner, before summarizing the results of the paper that this follows up on.*

As is detailed below in our response to reviewer #1, we restructured the Introduction along the lines suggested and reworded the review of previous work appropriately.

*3) The figures could be improved by including more labels and by providing a better connection to underlying measurements. At present some of the results are too derived and the connection to the measured data is unclear. The comparison with the previous T4 results could also be improved by showing some of the T4 and T5 data side by side.*

Again, as detailed below, we improved the manuscript by starting with a figure that shows the two models and their predictions, with individual flicker responses, sequence response and linear prediction, and extraction of the nonlinear response component. In Figure 2, we first show one example with the same sequence of raw and derived responses, using the same color code as in Figure 1. This should facilitate the interpretation of how we came up with the derived signals shown further on.

Furthermore, as suggested, we replot data from our previous report using identical color code to allow for a direct comparison of the T5 cells with the T4 cells.

*4) The Discussion could be improved by clarifying the key testable predictions made from the present work.*

We added and extended the previous Discussion in such a way that key testable predictions are listed and spelled out clearly.

*Reviewer #1:*
*This is a follow up on a paper from a year ago (Haag et al.) describing the presence of both preferred and null direction mechanisms for direction sensitivity in one type of fly T4 cell. The present paper extends this analysis, using an identical approach, to the other T4 cells and to T5 cells. I am not expert in fly vision, and thus cannot place this work in the context of the literature. The paper does not provide fundamentally new results, but shows that the results from Haag et al. apply more broadly to the T4 and T5 cells. The presentation of the work can be improved in several ways:*
*1) Introduction*
*The Introduction is not well suited for a broad audience. It starts with a long paragraph that almost immediately jumps into the anatomy of the fly motion circuits, but does not provide enough detail about the anatomy for someone unfamiliar with the organization of the fly visual system. It is, for example, hard to generate a clear picture from the introduction about how the circuitry for T4 and T5 cells differs. I think the Barlow-Levick and Hassenstein-Reichardt models, and the lack of direct tests of them, could be introduced earlier to make the central question of the paper clearer. Then maybe the anatomical information could be in a separate paragraph, once it has been clarified that you are interested in both excitatory and inhibitory circuits. Related to above, the second paragraph is clearly critical to the paper. I would consider adding a schematic to Figure 1 that shows the two models and the logic of the paired flash tests. I think that would go a long way to setting the paper up for a general reader.*

We agree with the reviewer and restructured the Introduction as suggested. While the first part now introduces the two models and their different types of predictions for apparent motion stimuli, the second paragraph introduces the fly visual system. This then is followed by a summary of the present literature, emphasizing the discrepancies and ending with a description of our previous results (Haag et al., 2016). In addition, we followed the suggestion by adding a schematic illustration of both models and the logic of the paired flash tests, including the data analysis as is used later on the experimental data. We hope that using the identical color code helps the reader to immediately assess what is done and how.

2) Comparison with past work.*The third paragraph of the Introduction describes the diversity of findings about how direction is computed. I think this could be done in a more objective manner. At present, the paragraph first presents the results and logic of the Haag et al. paper in some detail, and then presents findings of other work only in this context. Further, the language used to describe the other work is not objective (e.g. "but blamed the latter.…"). I think a better way to present this would be to say that there are discrepancies in existing studies, describe those different findings, and then describe the Haag et al. results. This would naturally set up the present work at the end of the paragraph.*

We followed the reviewer’s suggestion and reworded some of the sentences. We furthermore mention our previous study (Haag et al, 2016) in the end of the Introduction to make the transition to the present study (see above).

3) Figure 1.

*Figure 1 do not convey the nature of the underlying data and do not provide a sense of the reliability of the findings. First, some examples of the raw traces for a few selected locations would be helpful. Second, making the color schemes comparable between A and B is important since you are asking the reader to compare the two. Third, some quantification of the results with error bars is needed (the first paragraph of the Results section compares T4 and T5 but very qualitatively). Fourth, showing at least one example of the traces to each spot separately and together prior to subtraction would help a reader appreciate both Figure 1 and the differences between those and 1F. Fifth, some statistical comparison is needed for the comparisons in Figure 1 (see Results, third paragraph).*

We followed all suggestions of the reviewer and show raw traces as well as statistical analysis in the supplemental Figure 1. We furthermore show individual flicker responses, the responses to the sequence with the linear prediction and the resulting difference as the nonlinear response component in Figure 2.

4) Single vs. multiple columns.*The larger spots should activate multiple columns. Is this essential for T5 activation? This could get mentioned when it first comes up and the text could more clearly state that. For example, you say "from 1 to 10 degrees. Since stimuli delivered to the central column[…]" I would say "stimuli centered on the central column" since they are not restricted to one column. Generally this issue could be described and discussed more clearly.*

We agree with the reviewer that this issue was dealt with in a too compact way in the original submission. For the following, it is important to note that the spatial acuity of the fly eye is in the range of 5 degree. This means that any stimulus is spatially low-pass filtered by a Gaussian with 5 degrees full width at half maximum. Accordingly, enlarging the spot size will have two different effects: first, it will lead to an increasing peak intensity at the column where the stimulus spot is centered on, and second, it will lead to an increasing activation of neurons in neighboring columns (see Author response image 1). We now deal with this issue in the Discussion part of the revised paper. We now write: ‘One difference between T4 and T5 cells found in this study relates to the dependence of the directional motion signal on the spot size in the apparent motion paradigm (Figure 2). […] Which of these two effects is responsible for the higher threshold of T5 cells compared to T4 cells, and whether the sensitivity difference is in the input neurons or in T4 / T5 cells themselves, cannot be decided by the present study. ‘

Both these effects are shown in Author response image 1.

*Reviewer #2:*
*[…] 1) This study often refers to specific results and figure panels from the 2016 study, but comparisons would be made significantly easier if the results were presented side-by-side with the same color maps/scaling, whether this is done as part of supplementary figures or by repeating previous Figure panels in the new color maps.*

We agree with the reviewer and think this is a great idea. We therefore replotted the data from Haag et al., 2016 and show the T4 receptive field properties in Figure 1 side-by-side to the new T5 data, using identical color code.

*2) The authors discuss the differences in stimulus-size dependence between T4 and T5 cell responses (Figure 1). However, Figure 1 seems to show a significant difference between the stimulus-size dependence of ND suppression and PD enhancement within T5 cells. How do the authors interpret this?*

This is an important point: both T4 as well as T5 cells show a significantly lower threshold for null direction suppression than for preferred direction enhancement. We address this point at the end of the Discussion in the context of a possible biophysical implementation.

*3) In Discussion, the authors discuss at length the issue of neural implementation and mechanistic underpinnings for their model/results. For some part of this discussion, they refer to EM reconstructions of different medulla inputs to T4/T5 cells to potentially underlie their findings of PD enhancement on the null side of T4/T5 RFs and ND suppression on the preferred side of the RFs. With T5 cells in particular, they propose specific layouts of inputs and delays (and filtering) that would be consistent with their findings. Given that their study relies on calcium imaging and normalized responses to infer ND suppression, however, there is still some room for uncertainty about interpretation and mechanism. Thus, it would be helpful to readers to know what key properties of inputs they would consider inconsistent with the proposed spatially segregated combination of HR and BL type mechanism that their evidence suggests underlies T4/T5 responses (particularly T5, since there is some uncertainty about spatial location of inputs in this case, even from EM). Essentially, what would falsify their current model of direction selectivity? For example, does the model rely on some of the synaptic inputs to be inhibitory to produce ND suppression, or is that not a constraint? Essentially, a more in-depth discussion of caveats and laying out what key experiments would test these models may help better ground some of the interesting and intense debates in the field.*

Following the reviewer’s advice, we expanded our Discussion making clear predictions a) on the specific effects of individual blocking experiments, and b) on the specific synaptic polarity of each input neuron, i.e. whether it is expected to be excitatory or inhibitory. With respect to the transmitter phenotypes, recent studies seem to support our ideas for the T4 input neurons.

*Reviewer #3:*
*[…] Overall, the experiments are carefully done and the results seem fairly unambiguous. However, given that such a circuit organization was described by this group last year for the T4 cells (Haag et al., eLife, 2016), I'm having a hard time appreciating the major conceptual advancement to be made from the current work.*
*Demonstrating similar circuit mechanism for DS exists for the OFF system is undoubtedly an important step, but in and of itself it might appear as a detail for the general reader. The relevance of having two mechanisms for DS in different parts of the RF, and/or how the more subtle differences observed between ON and OFF pathways (e.g. Figure 1: size) work to improve DS coding for more complex stimuli (with dark/ and light), are important questions that have not been addressed.*

Our last paper showed the existence of both enhancement and suppression to be at work in T4 cells. However, our previous work was done on one subtype of T4 cells only. Moreover, given the conflicting evidence in the literature plus the different input neurons of T4 and T5 cells with quite different temporal dynamics (Arenz et al., 2017), identical mechanisms were not necessarily expected to account for direction selectivity in T4 and T5 cells. In our opinion, the present work, therefore, represents an important advance. We have tried to address the functional consequences of these results in the Discussion.

*Other comments:*
*Recently, orientation selective inhibitory field was described to sharpen DS in fly neurons by the Clandinin's group. It would be useful to discuss the reasons for the different results/model presented here.*

The results reported in Fisher et al. (2015) corroborated our previous results from Maisak et al. (2013) where we showed that ‘gratings flickering in counter-phase lead to layer-specific responses, depending on the orientation of the grating (Supplementary Figure 1 of Maisak et al., 2013)’. This is clearly spelled out in Fisher et al. (2015): ‘These results are consistent with previous observations in which the responses of T4 and T5 cells to counterphase flicker depended on the orientation of the stimulus (Maisak et al., 2013)’. In addition, these results can be explained by the experiments shown in Figure 1. For T4/T5-cells in layer 3 a vertically extended bar would lead to a reduced flicker response due to the inhibition. Thus, we see no discrepancies concerning the orientation tuning of T4 and T5 cells.

*It is also important to discuss the anatomical arrangement of the Mi4 and Mi9 cells that are known to inhibit T4s (i.e. are they displaced to the preferred side?)*

We addressed the anatomical arrangement in the discussion by writing: ‘Columnar input neurons contact the T4 cell dendrite in a way that depends on the direction tuning subtype: while Mi9 synapses are clustered on the null side of the dendrite, Mi1 and Tm3 synapse on the central part and Mi4 are found predominantly on the preferred side.’. Data from Takemura et al. (2017) suggest Mi9 to be glutamatergic, and Mi4 to be GABAergic. Given the expression of the GluCl channel in different neurons of *Drosophila*, both Mi9 and Mi4 are potentially inhibitory. The OFF-center property of Mi9 would thus be turned around, making its effect on the T4 cell positive (release from inhibition). However, all this is speculative at the moment since no one has directly demonstrated an inhibitory effect of neither Mi4 nor Mi9 on the membrane potential of T4 cells.

*The nomenclature is confusing. Usually, the preferred side refers to the side at which the preferred stimulus enters a DS neurons receptive field. The preferred and null sides have not been defined and appear to be opposite to the normal convention.*

The reviewer is correct in both points: first, we have missed to define the ‘preferred’ and the ‘null side’, and second, we have used these terms opposite to the normal convention (see e.g. Briggman et al., 2011: ‘Dual whole-cell recordings have shown inhibitory connections between SACs and DSGCs to be stronger when the SAC’s soma is located on the null side (that is, the side from which null-direction stimuli approach) of the DSGC’s soma.’. We apologize for that. In our revised version, we now define and use these terms following the normal convention.

*Figure 4. The DS tuning appears to be extremely sharp which makes it challenging to understand how 'in between' directions (e.g. 60,120, 210, 315 degrees) would be computed.*

The polar histogram indeed reveals a highly peaked distribution. This histogram, however, does not represent the directional tuning but rather the distribution of preferred directions, i.e. at which angle the cells respond maximally. The direction tuning, i.e. the responses of a given cell as a function of the direction of the moving pattern is in fact broader leading to about 50% half-maximum responses to grating motion at 45 degrees (see Author response image 2, taken from Haag et al., 2016, Figure 6B).

**Author response image 2. respfig2:** 

*The nonlinear components for increased null direction inhibition or increased preferred direction excitation appear to have different time courses. In many cases (especially in Figure 1, but maybe it's less stereotyped in Figure 2–Figure 3), the increases in ND inhibition occur earlier and decay faster than the increases in PD excitation. Were any experiments done investigating the relative time course of the two mechanisms? It would be useful to discuss some possible reasons for why there are temporal differences between the two mechanisms.*

This is an interesting observation and we thank the reviewer for pointing this out to us. In response to this comment, we have now averaged the time courses of the preferred direction enhancement and null direction suppression of all four subtypes of T4 and T5 cells. To allow for a better comparison, we have flipped the null direction suppression. The resulting data, as shown in Author response image 3, do not indicate any significant differences between the two time courses of the nonlinear response components for preferred and null direction, neither in T4 nor in T5 cells.

**Author response image 3. respfig3:** 

[Editors' note: further revisions were requested prior to acceptance, as described below.]

The revisions did a good job addressing most of the comments raised in review. One substantive issue remains: a need for statistical analyses to accompany the conclusions reached from Figure 1 and 2. This should particularly include the comparisons of T4 and T5 cells.

As for Figure 1, we added a supplemental figure showing the data for Figure 1 as a bar histogram representing the mean ± SEM. The main conclusion that we draw from the experiments shown in Figure 1 and is the inhibition of the responses in the dorsal receptive field. In order to proof the significance of the effect, we calculated the p-values for the differences in the responses for the three dorsal ommatidia (number 19, 8, 9 in the scheme) and three ventral ommatidia (number 13, 14, 15) using a two-sided t-test. The values are given in the figure legend of the supplemental figure.

In addition, we now performed two-sided t-tests for the data shown in Figure 2 and F. Taking a p-value of 0.05 we marked the significant differences between T4-cells and T5-cells for preferred and null-direction responses with asterisks.